# SHADOW: Leveraging Segmentation Masks for Cross-Embodiment Policy Transfer

**Marion Lepert**[1]     **Ria Doshi**[2]     **Jeannette Bohg**[1]

[1]Stanford University     [2]UC Berkeley

**Abstract:** Data collection in robotics is spread across diverse hardware, and this variation will increase as new hardware is developed. Effective use of this growing body of data requires methods capable of learning from diverse robot embodiments. We consider the setting of training a policy using expert trajectories from a single robot arm (the *source*), and evaluating on a different robot arm for which no data was collected (the *target*). We present a data editing scheme termed **Shadow**, in which the robot during training and evaluation is replaced with a composite segmentation mask of the source and target robots. In this way, the input data distribution at train and test time match closely, enabling robust policy transfer to the new unseen robot while being far more data efficient than approaches that require co-training on large amounts of data from diverse embodiments. We demonstrate that an approach as simple as Shadow is effective both in simulation on varying tasks and robots, and on real robot hardware, where Shadow demonstrates an average of over 2x improvement in success rate compared to the strongest baseline. Videos are available at https://shadow-cross-embodiment.github.io.

## 1   Introduction

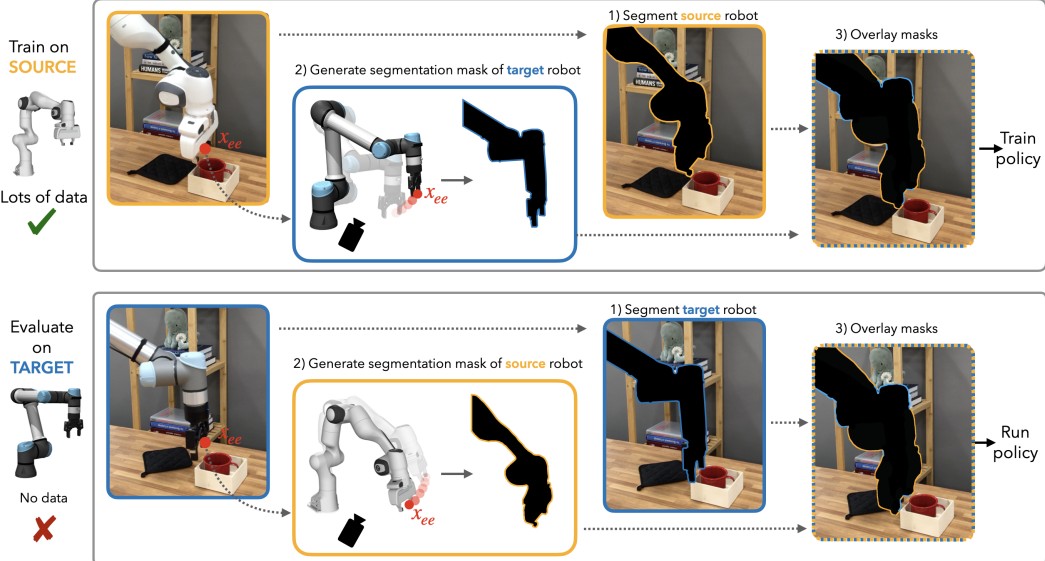

Figure 1: Schematic of **Shadow**. Policy is trained on source robot and evaluated on the target robot. No data is collected for the target robot. Observation images are overlaid with the segmentation masks of the source and the target robots. During training, the target robot is rendered to be at the same end-effector pose, $x_{ee}$, as the source robot and vice-versa during evaluation.

8th Conference on Robot Learning (CoRL 2024), Munich, Germany.

Increasing the size of open-sourced robotics data is crucial to advancing robotics research. The last several years have seen exciting large-scale collaborative robot data collection efforts in an effort to address this need [1, 2, 3, 4]. However, many of these efforts are spread across various robot hardware, and this variation is likely to increase as new hardware is developed and more contributors share their data. Therefore, effective use of this limited but growing body of data requires methods capable of learning from diverse robot embodiments.

As hardware improves, it is unrealistic to require collecting large amounts of data on every new embodiment. Therefore, we need methods that can generalize to new embodiments without the need for large-scale embodiment-specific data collection.

One of the main reasons why policy learning across different embodiments is challenging is that robots look visually quite different (Figure 2) and vision based policies suffer dramatically when visual observations are disturbed. The number of different robot embodiments for manipulation in use by research labs today remains quite low – 18 as seen in OXE [3] – and given current algorithms, is far too little to provide enough visual diversity to generalize policies to unseen embodiments.

We consider the setting of training a policy using expert trajectories from a single robot arm, and evaluating on a different robot arm for which no data was collected. We present a data editing scheme termed **Shadow** (Figure 1). At train time, the source robot is replaced by a composite segmentation mask: a mask of the source robot overlaid with a mask of the target robot such that its end effector pose matches that of the source robot. At test time, the converse data edit is performed: the target robot is replaced by a composite segmentation mask of the target robot overlaid with a mask of the source robot such that its end effector pose matches that of the target robot. The segmentation mask of either robot in the target pose can be easily generated since current joint angles, the kinematics, and CAD models of each robot are known.

The principal advantage of Shadow is that the input data distribution at train time and test time match closely. In either case, the "active" robot is replaced with a composite segmentation mask of the source and target robots. In contrast to other image augmentation and in-painting methods, Shadow deliberately discards the RGB details of the robots (which we show to be unimportant – see A.3). Instead, Shadow preserves the crucial gripper/object interaction, and the RGB details of the surroundings (which can be distorted with prior in-painting methods [5]). Additionally, compared to methods that co-train on large datasets with diverse embodiments, our method is significantly more data efficient: it does not require any data to be collected on the target robot; and is able to generalize to an unseen target robot from data collected on only a single source robot. Finally, our method does not require an image of the scene with no robot, an assumption required by methods such as keypoint tracking [6] and affordance prediction [7].

**To summarize: our main contribution is Shadow, an efficient data editing scheme for robust cross-embodiment learning that uses composite segmentation masks to train a policy using data collected on a source robot to transfer to an unseen target robot.**

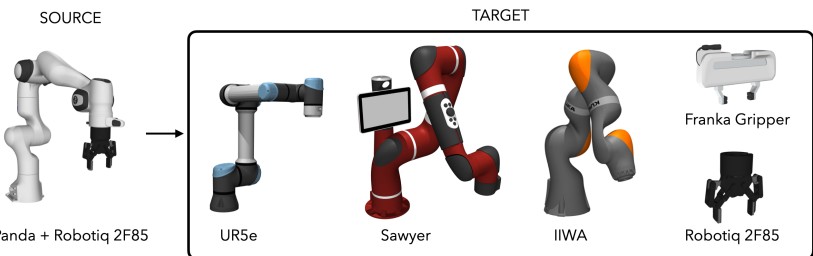

Figure 2: In simulation, we evaluate the effectiveness of Shadow in transferring a policy trained on a Panda with a Robotiq 2F85 gripper to the target robot/gripper combinations shown here. On real-world hardware, we transfer to the Panda robot with the Franka gripper and the UR5e robot with the Robotiq 2F85 gripper.

## 2 Related Works

### 2.1 Cross-Embodiment Transfer

To improve policy transfer, prior work has utilized wrist cameras to minimize visual differences in the input observation space [8]. However, these works limit tasks to those that do not require a thorough view of the environment and make it impossible to leverage the valuable amount of existing robot data collected from third-person viewpoints. Some works [9] mask robot pixels, train a video prediction model that predicts non-robot pixels, and use visual foresight when rolling out the policy. Such video prediction models are compute-intensive. On the architecture side, Furuta et. al. [10] trains a task and morphology specific policy, distilling each combination into a transformer. Similarly, Devin et. al [11] train a separate head for each robot and task. Other work [10, 12, 13, 14] follows this line of work assuming a known distribution of robots at eval-time. Another subset of multi-robot works condition on morphology-specific information [15, 16, 17, 18], or train a latent vector for a new robot from few shot trajectories. These works all require data collected from the target robot prior to evaluation, whereas our method aims to perform on an entirely new robot without any collected trajectories.

### 2.2 Large-Scale Multi-Embodiment Datasets

A number of recent works in robot learning pre-train policies on large, diverse robotic datasets [19, 20, 21, 22, 23, 24] as a means of generalizing to new embodiments. Some works use Cartesian space actions [3, 25], but cannot be deployed zero-shot to a new robot because they do not leverage any prior knowledge of the robot's dynamics, such as a robot's URDF, nor do they align action spaces. Our method allows us to take a policy trained on a single robot and generalize to any single-arm robot. Other prior work [26] uses human demonstrations to extract object-centric trajectories that the target robot can use to collect its own demonstrations. However, we do not require the collection of target robot demonstrations nor do we impose object-centric constraints on our tasks.

### 2.3 Image Augmentation and In-painting

Image editing as a form of domain randomization to build more robust policies is another approach to generalization [27, 28, 29]. SuSIE [30] queries a diffusion model for subgoal generation for goal-conditioned policies. Another work [31] simulates different robot viewpoints by augmenting existing data. Video in-painting [32] has been used to mask out human hands and replace them with robot grippers. These works rely on large diffusion models which are computationally expensive to train or fall short in instances where the delineation between the gripper and object must be very precise, as in-painting models can suffer from blurriness. Probably most similar to our work is Mirage [5], which trains a policy on a source robot and inpaints the target, unseen robot over the source robot during evaluation time. In this way, no data is required for the downstream, target robot. However, this method also inherits the short-comings of in-painting models, limiting generalizability to more visually-complex backgrounds or settings where the source and target robot are drastically different (e.g., inpainting a WidowX over a Franka robot).

## 3 Approach

Our method, **Shadow**, enables training a policy on one robot embodiment and deploying it on another robot embodiment without needing to collect any new data on the second robot embodiment. Our key insight is that generating segmentation masks of robot embodiments at a desired end-effector pose is easy and fast given access to a robot's geometry, joint angles and a well-calibrated camera.

## 3.1 Problem Setup

We consider the scenario where we have access to a dataset $D = \left\{ \left( o_1^i, s_1^i, a_1^i, \ldots, o_{H^i}^i, s_{H^i}^i, a_{H^i}^i \right) \right\}_{i=1}^N$ of $N$ trajectories on a given robot embodiment, referred to as the *source* robot. We use the term embodiment to mean a *type* of robot such as the Franka, Kinova, IIWA, etc. Each observation $o_t := (I_{t-T}, .., I_t)$ is a stack of RGB images from the $T$ previous steps and each state $s_t$ corresponds to the joint positions of the robot. Each action $a_t$ is the absolute Cartesian pose of the end-effector. We evaluate our method on a different robot embodiment, referred to as the *target* robot, for which no trajectories are collected. The pose of the camera is calibrated relative to the base of both robots, resulting in known transformations $\mathcal{T}_{S_0}^{\text{Camera}}$ and $\mathcal{T}_{T_0}^{\text{Camera}}$, where $S_0$ and $T_0$ are the base frames of the source and target robots, respectively. While our setup is limited by the need for camera calibration parameters, this information is frequently collected in open-sourced robot datasets ($46\%$ of Open-X Embodiment [3] data).

Our method does not require that the pose of the bases of the source and target robots be the same, but we assume that the distribution of locations of the objects to be manipulated relative to the camera stays the same. We assume that the background, lighting, and pose of the camera in the scene stays fixed for the two robots, as robustness to scene variations is outside this paper's scope.

Different embodiments cause significant distribution shift in the image observation space. We focus on how to train vision-based policies to be robust to this distribution shift and therefore only consider tasks that both the source and target robot embodiments can complete using similar strategies.

## 3.2 Shadow data editing

### 3.2.1 Training

For each image in our training dataset, $I_t$, we use robot proprioception, known kinematics and geometry of the robot, and camera parameters to segment out the pixels corresponding to the source robot and set them to a fixed color, $c$. While $c$ can be any color, we use solid black for simplicity. We refer to this modified image as $I_{t,*}$.

In order to generate a segmentation mask of the target robot, first we model a virtual rendering of the target robot with its end-effector in the same Cartesian pose as the end-effector of the source robot. The joint positions of the target robot can be solved using any robot controller; in our work, we use operational space control [33]. We render the target robot image using a simulated camera with the same intrinsics and extrinsics, $\mathcal{T}_{T_0}^{\text{Camera}}$, as the real world camera. We then extract the segmentation mask of the target robot from this image, and overlay the mask (with all its pixels set to $c$) onto $I_{t,*}$ to obtain $I_{t,\bullet}$. If the target robot has a different gripper embodiment, we also overlay the target gripper segmentation mask set to $c$. For clarity, $I_{t,\bullet}$ is $I_t$ modified to have the source robot "blacked out" with its segmentation mask, and the target robot mask further super-imposed on the image. We then train a policy with imitation learning on the modified dataset $D_\bullet = \left\{ \left( o_{1,\bullet}^i, a_1^i, \ldots, o_{H^i,\bullet}^i, a_{H^i}^i \right) \right\}_{i=1}^N$.

### 3.2.2 Evaluation

To evaluate our policy on the target robot, we follow a similar process as done during training. We take the current image $I_t$ of the scene and segment out the pixels corresponding to the target robot and set them to the fixed color $c$, obtaining $I_{t,\star}$. We generate a virtual rendering of the source robot with its end-effector in the same Cartesian pose as that of the target robot. We render the source robot image using a simulated camera with the same intrinsics and extrinsics, $\mathcal{T}_{S_0}^{\text{Camera}}$, as the real world camera. We then extract the segmentation mask of the target robot from this image, and overlay the mask onto $I_{t,\star}$ to obtain $I_{t,\bullet}$. This gives us $o_{t,\bullet}$ which forms the input to our policy at test time to get the next action. We execute the action using a blocking controller for accurate trajectory tracking.

Because we have access to an exact CAD model of the robot, the segmentation masks of the robots generated from virtual renderings have no loss of fidelity beyond noise introduced from camera calibration error. Therefore, in a properly camera calibrated setup, we expect minimal distribution

shift between the images $I_{t,\bullet}$ in our training set and those obtained during evaluation on either the source or target robot, enabling robust cross-embodiment policy transfer.

## 4 Results

We hypothesize that Shadow enables policy transfer to unseen target robots with minimal degradation compared to a policy trained and evaluated on raw source robot images. To test this, we evaluate our method in simulation and on real hardware, using Diffusion Policy for imitation learning [34].

### 4.1 Baselines

We compare our method to a naive baseline: training on the unmodified source robot data and evaluating on unmodified images of the target robot. We also compare to the closest method to ours, Mirage [5], a cross-embodiment method that leverages in-painting to transfer between source and target robots. We re-implement Mirage to use Diffusion Policy instead of the LSTM behavior cloning policy used in the original paper for a fair comparison to our method. We do not explicitly compare to a large model pre-trained on diverse embodiments (e.g. Octo model [25], which is pre-trained on a mixture of 25 datasets from OXE), because Mirage was already reported to outperform Octo by 20-50 percentage points for cross-embodiment learning on similar tasks [5].

We also compare to a model that is an upper-bound of performance on a given task: a model trained using raw images of the source robot and evaluated on the source robot (i.e., no data editing at train or test time, see A.7). In each results figure, the dashed line denotes performance of this model. Asterisks denote methods that are statistically inferior to Shadow (two-proportion z-test, $\alpha = 0.05$).

### 4.2 Simulation experiments

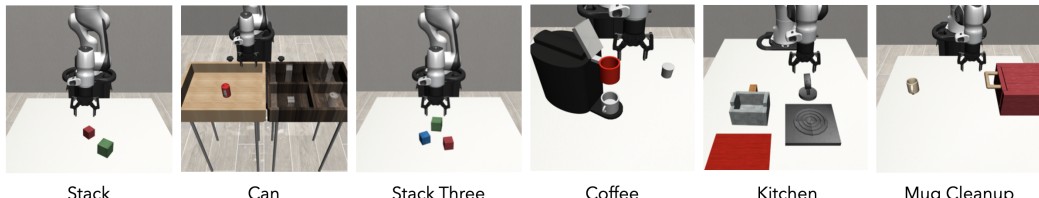

Figure 3: The six tasks evaluated in simulation, shown from the viewpoint used during training.

We evaluate our method in the Mujoco [35] simulator using the Mimicgen [36] and Robomimic [37] datasets and the robot models from [38]. We use the Panda with the Robotiq 2F-85 gripper as our source robot and evaluate performance on the Sawyer, IIWA, and UR5e target robots with two grippers across six tasks (Figure 3). For each task, we use 190-950 demos on the source robot, depending on the amount of available data in Mimicgen and Robomimic for the source robot (A.5).

We compare the success rates between each method, task, and embodiment combination over 100 trials each (Figure 4, 5). The performance of the naive baseline is close to $0\%$ on all tasks and embodiments, indicating that the distribution shift caused by the change in embodiment drastically decreases performance even on the simplest stack task. The Mirage baseline performs well on the easiest tasks - Stack and Can - but its performance degrades on tasks that require higher precision such as the Stack Three task which requires balancing three objects on top of each other or the Coffee task which requires inserting a coffee pod into a narrow slot. Mirage relies on in-painting which causes distortion in the image and generates erroneous artifacts which change the image distribution. In contrast, Shadow achieves performance on the target robot comparable to the performance on the source robot in all tasks except Mug cleanup, in which it still significantly outperforms both baselines. The Mug Cleanup task is especially difficult for the Franka gripper because the gripper can jam against the side of the drawer due to its large width, whereas the source gripper does not.

Note that Mirage struggles even when the robot stays the same but only the gripper changes (Figure 4 - Panda). The is because the distances from the flange of the robot to the end-effector control point are different between the Robotiq 2F-85 gripper and the Franka gripper. As a result, moving the different grippers to the same control point requires moving the flange of the robots to different locations. Mirage still requires the use of in-painting in this scenario, and Shadow still needs to overlay two segmentation masks. Therefore, even in simulation, changing only the gripper is challenging.

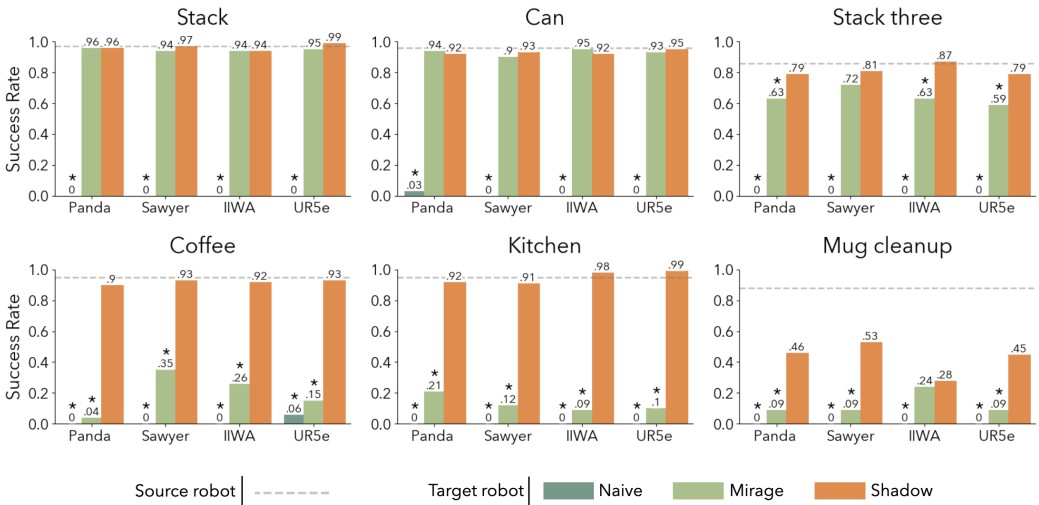

Figure 4: **Different robot, different gripper**: Evaluation in simulation over six tasks and four target robots (Panda, Sawyer, IIWA, UR5e; each with the Franka gripper) (100 roll-outs per evaluation). The source robot is Panda+Robotiq 2F85 gripper. Dashed line: policy trained and evaluated on source robot. ∗ denotes statistically inferior to Shadow ($p < 0.05$). Shadow outperforms Mirage on all tasks, and shows no performance degradation compared to source robot in 5/6 tasks.

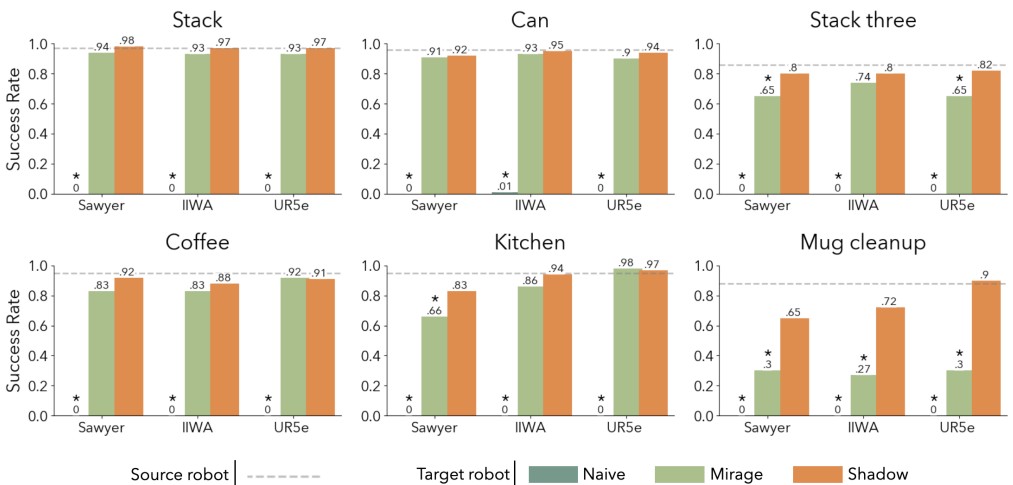

Figure 5: **Different robot, same gripper**: Evaluation in simulation over six tasks and three target robots (Sawyer, IIWA, UR5e; each with the Robotiq 2F85 gripper) (100 roll-outs per evaluation). The source robot is Panda+Robotiq 2F85 gripper. Dashed line: policy trained and evaluated on source robot. ∗ denotes statistically inferior to Shadow ($p < 0.05$). Shadow outperforms Mirage on all tasks, and shows no performance degradation compared to the source robot in 5/6 tasks.

|              | Panda (Source) | UR5e (Target) | IIWA (Target) |
| ------------ | -------------- | ------------- | ------------- |
| Black-only   | 0.94           | 0.02          | 0             |
| Shadow       | 0.98           | 0.97          | 0.97          |

Table 1: Ablation of target mask in sim on the Stack task. (100 roll-outs per evaluation)

The Shadow data editing procedure during training is the union of two changes to the raw image: (1) the source robot is "blacked out", (2) the mask of the target robot mask is overlaid. To evaluate the respective contribution of these two separable edits, we ablate (2) and consider a data editing procedure that only blacks out the source robot (Table 5). "Black-only" does not enable policy transfer to target robots UR5e or IIWA at all, while Shadow results in near-perfect performance. This demonstrates that the overlay of the target robot mask is crucial for the efficacy of Shadow.

### 4.3 Real world experiments

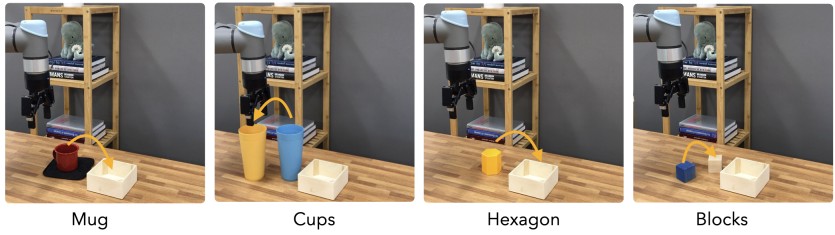

Figure 6: The 4 tasks evaluated in real-world experiments. Mug and Hexagon require placing the respective objects in the wooden container. Cups and Blocks require stacking the respective objects.

In the real world, we use the Panda robot with the Robotiq gripper as the source robot and collect 300 demos per task. We consider two cross-embodiment scenarios. First, we keep the same robot, but change the Robotiq gripper to a Franka gripper. As explained above, this scenario is challenging because the distance to the end-effector control point is different between the grippers, meaning that the configuration of the Panda robot will be different between the source and target scenarios for the same action. Second, we keep the same gripper, but change the Panda robot to the UR5e robot. We compare Mirage and Shadow on each task and embodiment combination using 50 trials each. For hardware safety, we do not evaluate the naive baseline since it performs poorly in simulation.

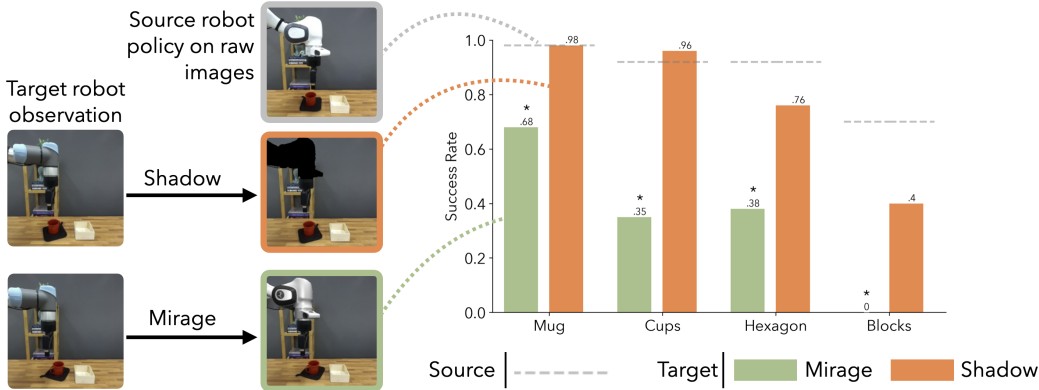

Figure 7: **Different robot, same gripper**: Real-world evaluation over four tasks. The target robot is UR5e + Robotiq 2F85 gripper, and the source robot is Panda + Robotiq 2F85 gripper. (50 roll-outs per evaluation) $*$ denotes statistically inferior to Shadow ($p < 0.05$). Shadow outperforms Mirage baseline on all tasks, and shows no performance degradation compared to source robot in 2/4 tasks.

Our results demonstrate that Shadow significantly outperforms Mirage in all real world tasks. We notice that the primary failure mode of Mirage is that it sometimes executes seemingly random actions that are irrelevant to the task, which we hypothesize is due to the distortion of the image caused by in-painting. In contrast, Shadow's primary failure mode is lack of precision, where it sometimes slightly misses a grasp and is not able to recover. Performance is lowest on the Hexagon and Blocks tasks because they require grasping wider objects which means the robots must precisely position themselves above the objects before closing the gripper. In contrast, a wider range of gripper positions will result in a successful grasp of the narrow rim of the cup or mug's edge.

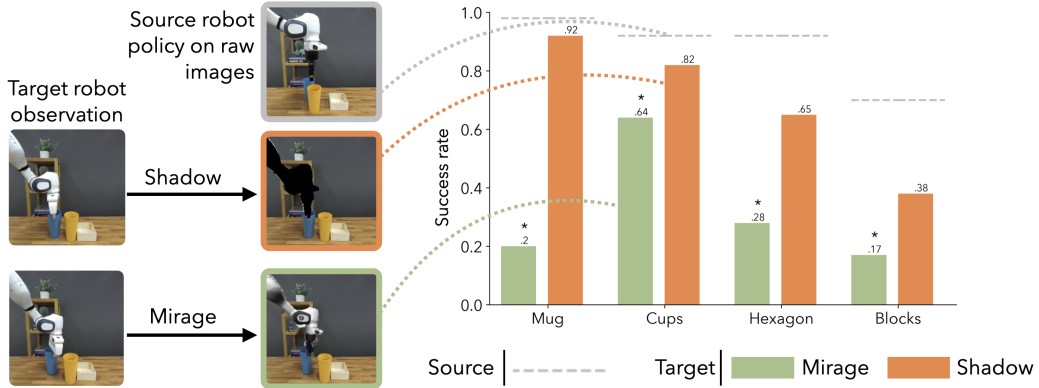

Figure 8: **Same robot, different gripper**: Real-world evaluation over four tasks. The target robot is Panda + Franka gripper, and the source robot is Panda + Robotiq 2F85 gripper. (50 roll-outs per evaluation) $*$ denotes statistically inferior to Shadow ($p < 0.05$). Shadow outperforms Mirage baseline on all tasks.

## 5  Limitations

In order to correctly generate segmentation masks of the source and target robots, Shadow relies on accurate camera calibration parameters and accurate proprioception information. Additionally, Shadow requires a policy to be trained for every new target robot embodiment, although data from a single source is sufficient for any number of targets. In situations where the robot is partially occluded by objects in the scene, the overlaid segmentation mask will obfuscate these objects. This can be addressed with depth data, but may be limited by the depth data quality. Lastly, Shadow currently does not generalize to novel scenes, but it can be readily combined with orthogonal work on inducing robustness to scene variations to address this limitation. See A.1 for more details.

## 6  Conclusion

We propose Shadow, an effective method for cross-embodiment policy transfer. We show in simulation and on real hardware that we can train a policy on a dataset of trajectories collected on one robot and obtain robust performance on another robot for which no data has ever been collected. The main advantages of our approach are its simplicity, its close alignment of the train and test set distributions despite different embodiments, and its data efficiency compared to co-training approaches.

There are many avenues for future work. First, while Shadow is theoretically compatible with target robot embodiments placed anywhere in a scene, further investigation is needed to assess its robustness to different robot placements. Second, although we demonstrated Shadow's robustness across various target robots, each trained policy was specific to a source/target robot pair. Future work could explore Shadow's ability to allow a single policy to generalize across multiple embodiments. Finally, improving robustness to camera calibration errors via data augmentation provides another clear opportunity for improvement.

**Acknowledgments**

This work is supported by the National Science Foundation under Grant Number 2327974. We thank Connor Yako and Ken Salisbury for letting us borrow their UR5e robot, Lawrence Yunliang Chen for helpful responses to our questions on Mirage, and Claire Chen for help with physical experiments.

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

# A Appendix

## A.1 Limitations

1. **Camera calibration noise**: Shadow relies on accurate camera calibration in order to generate segmentation masks of the source and target robots. We investigate the robustness to camera calibration noise in Section A.4 and find Shadow deteriorates proportional to the scale of noise added. In future work, we could attempt to render Shadow more robust to camera calibration error via (modest) noise injection during training.

2. **Training is linear with number of embodiments**: We demonstrated that Shadow performs well on various target robots, but each individual policy we trained was for a specific source/target robot pair. Note, however, that training an additional policy for a new target does not require any additional data to be collected (i.e., data from a single source is sufficient for any number of targets). In future work, we also hope to evaluate Shadow's ability to allow a single policy to generalize over more than two embodiments. Specifically, we could try overlaying segmentation masks of two or more target robot embodiments and evaluating if Shadow continues to perform well.

3. **Occlusions**: In situations where the robot is (partially) occluded by objects in the scene, the overlaid segmentation mask will obfuscate these objects. As a result, the model may have difficulty disambiguating between cases where the robot is in front of an object versus behind it. In simulation, this can be easily addressed by using depth information from the original scene to only mask robot pixels behind occluding objects. In real, this can be addressed with real-time depth data (either by using a stereo camera or monocular depth estimation), but may be limited by the depth data quality. Additionally, the source and target robots cannot be positioned in such a way that their mask overlay occludes a significant part of the scene such that the model can no longer see the task at hand.

4. **No generalization to new scenes (yet)**: For each task, data is collected on the source robot in a single scene, without adding any diversity to the background elements of the scene. As a result, Shadow cannot generalize to novel scenes on either the source robot or the target robot. However, our method can be used in combination with orthogonal work on inducing scene robustness (e.g. collecting data in different scenes with different distractor objects, lighting conditions and/or doing data augmentation) to enable Shadow to work in scenarios where the source and target robot are in different scenes.

5. **Accurate proprioception**: Shadow relies on accurate measured joint angles and an accurate kinematic model of a robot in order to properly segment out robot pixels in an image. This may make adoption of Shadow on low-cost and less precise hardware more challenging.

6. **Embodiments rely on the same physical strategies**: The goal of Shadow is to train a policy to output the same end-effector actions regardless of the visual appearance of the robot embodiment. Therefore, Shadow works across embodiments and tasks where the same set of actions can be used to complete the task. Shadow is well suited to work with commonly used robots that have parallel jaw grippers and full 6-DOF end-effector control. However, the method may struggle in tasks with cluttered environments where embodiments might collide with the environment in different ways, requiring different physical strategies. Shadow is also not able to transfer between parallel jaw grippers and dexterous hands.

## A.2 Does Shadow perform well on the source robot?

The goal of Shadow is to develop a policy that performs well on the target robot despite never having collected data on the target robot. In this section, we investigate whether or not this same policy can also perform well on the source robot. In simulation, we run evaluations on the most challenging cross-embodiment scenario: different robot and different gripper. Results in Figure 9 demonstrate no significant change in performance between Shadow's performance on the source and target robots across the majority of the tasks. For the Mug Cleanup task, we see increased performance on the

source robot, which makes sense given that the target robot uses the Franka gripper which tends to jam against the drawer in the Mug Cleanup task whereas the source robot with the Robotiq gripper does not suffer from the same problem.

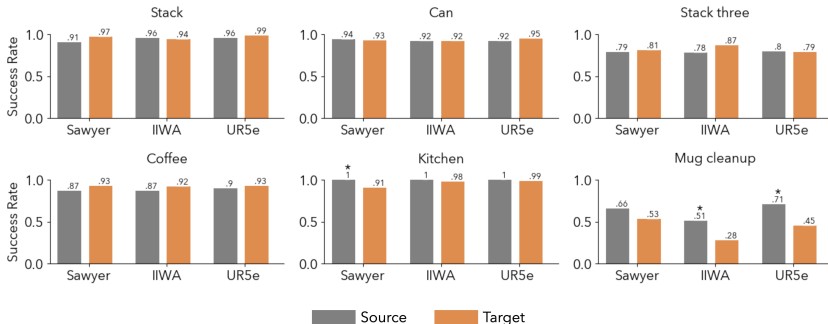

Figure 9: Shadow's performance on six simulation tasks. The source robot is the Panda + Robotiq 2F85 gripper and the target robots use the Franka gripper. Shadow performs equally well on the source and target robots across the majority of tasks. (100 roll-outs per evaluation).

### A.3 Does masking out robot pixels affect performance?

Shadow relies on masking out the robot's pixels in the observation images. Our intuition for the feasibility of this approach stemmed from an experiment we ran in simulation that showed that replacing a source robot with its segmentation mask (i.e., discarding all RGB information about the source robot) had no effect on task performance (Table 2, left). In other words, the RGB values of the robot itself are not important for completing the tasks studied. Note that this section is independent of cross-embodiment learning - we simply replaced the source robot with its segmentation mask.

We ran a similar experiment on real robot hardware and came to the same conclusion: on all tasks, replacing the robot with its segmentation mask during training and evaluation did not affect task performance (Table 2, right).

|  | Simulation | | | | | | Real world | | | |
|---|---|---|---|---|---|---|---|---|---|---|
|  | Stack | Can | Stack 3 | Coffee | Kitchen | Mug | Hex. | Cups | Stack | Mug |
| Vanilla | 0.97 | 0.96 | 0.86 | 0.95 | 0.95 | 0.88 | 0.92 | 0.92 | 0.7 | 0.98 |
| Masked | 0.99 | 0.96 | 0.86 | 0.93 | 1.0 | 0.84 | 0.89 | 0.92 | 0.76 | 1.0 |

Table 2: Replacing the source robot with its segmentation mask does not affect task performance in either simulation or on real robot hardware. 100 roll-outs per evaluation in simulation and 25 roll-outs per evaluation in real. Vanilla: policy trained/evaluated on raw RGB images, Masked: policy trained/evaluated on images with the source robot replaced by its mask. No comparisons between Vanilla and Masked for a given task are statistically significant.

### A.4 Sensitivity to camera calibration

Shadow relies on accurate camera calibration in order to generate segmentation masks of the source and target robots. We investigated the sensitivity of Shadow's performance on real hardware to additional noise introduced to camera calibration extrinsics at evaluation time on the target robot (Table 3). As expected, the performance of Shadow deteriorates proportional to the scale of noise added.

Note that we did not deliberately add any camera calibration noise during training. In future work, we could attempt to render Shadow more robust to camera calibration error via (modest) noise injection during training.

| $\Delta\mathbf{x}$ (m) | $\Delta\theta$ | Mug | Hexagon |
|---|---|---|---|
| 0 | 0° | 0.98 | 0.76 |
| 0.01 | 5° | 0.64 | 0.16 |
| 0.02 | 10° | 0.20 | 0 |

Table 3: Sensitivity of Shadow to camera calibration error. We evaluated the success rate on real hardware on the target robot in different noise conditions (source: Panda robot with Robotiq gripper, target: UR5e robot with Robotiq gripper). $\Delta\mathbf{x}$ and $\Delta\theta$ denote the standard deviation of 0-mean Gaussian linear and angular noise deliberately added to the camera extrinsic parameters during evaluation, respectively. (25 roll-outs per evaluation).

## A.5 Dataset generation details

To build our training datasets in simulation, we leverage existing datasets from Robomimic [37] and Mimicgen [36]. These datasets contain demonstrations for a Panda with a Franka gripper. We convert these demonstrations to our source embodiment, a Panda with a Robotiq gripper, by rolling out each demonstration from the original dataset onto our source embodiment. Due to small differences in robot dynamics and collisions with the scene, not all rollouts are successful. We report the final dataset sizes for each task in Table 4.

| Task | Dataset origin | Number of demos |
|---|---|---|
| Stack | Mimicgen | 951 |
| Can | Robomimic | 190 |
| Stack three | Mimicgen | 937 |
| Coffee | Mimicgen | 768 |
| Kitchen | Mimicgen | 863 |
| Mug Cleanup | Mimicgen | 660 |

Table 4: Source robot data used to train Shadow

For real world experiments, we collect 300 demonstrations per task on the source embodiment by teleoperating the robot with an Oculus controller.

## A.6 Policy implementation details

Table 5 describes the hyperparameters used to train Diffusion Policy. The same hyperparameters were used for all methods, and we used the CNN-based Diffusion Policy architecture. In simulation, all tasks used the DDPM scheduler with 100 training diffusion steps and 100 inference steps. In the real world, all tasks used the DDIM scheduler with 100 training diffusion steps and 10 inference steps. All hyperparameters not listed are unchanged from the original Diffusion Policy paper.

| | Task | To | Ta | ImgRes | Batch size | Lr | Epochs | Horizon |
|---|---|---|---|---|---|---|---|---|
| Real | Mug | 2 | 8 | 240 | 256 | 1e-4 | 2000 | 200 |
| | Cups | 2 | 8 | 240 | 256 | 1e-4 | 2000 | 200 |
| | Hexagon | 2 | 8 | 84 | 256 | 1e-4 | 2000 | 200 |
| | Blocks | 2 | 8 | 84 | 256 | 1e-4 | 2000 | 200 |
| Sim | Stack | 2 | 8 | 84 | 256 | 1e-4 | 5000 | 200 |
| | Can | 2 | 8 | 84 | 256 | 1e-4 | 5000 | 200 |
| | Stack Three | 2 | 8 | 84 | 256 | 1e-4 | 5000 | 400 |
| | Coffee | 2 | 8 | 84 | 256 | 1e-4 | 5000 | 300 |
| | Kitchen | 2 | 8 | 84 | 256 | 1e-4 | 5000 | 750 |
| | Mug Cleanup | 2 | 8 | 240 | 256 | 1e-4 | 5000 | 400 |

Table 5: Hyperparameters used for diffusion policy. **To**: observation horizon, **Ta**: action horizon, **Epochs**: Number of training epochs, **Horizon**: max number of rollout steps

Our method can run at up to 20 Hz (for 84x84 images; or 10 Hz for 240x240), but we execute our policy on the real hardware at only 2Hz due to hardware limitations. In the set of experiments testing out a different gripper, we are forced to reduce our control frequency because the max control frequency of the Franka gripper is only 2Hz. In the set of experiments testing out a different robot, we prioritized hardware safety, as we were borrowing the UR5e, and chose to use the UR5e's slow MoveL blocking command to send Cartesian end-effector pose commands because the MoveL command comes with inherent safety features. The MoveL blocking command forced us to slow our policy down to 2Hz.

### A.7 Upper bound of performance

For our main results, we compare our method to a model that is an upper-bound of performance on a given task: a model trained using raw images of the source robot and evaluated on the source robot (i.e., no data editing at train or test time, no cross-embodiment transfer). We use the source robot instead of the target robot as an upper bound of the target robot's performance because training on the target robot and then evaluating on the target robot would require collecting a new dataset of demonstrations on the target robot. Because real world data is collected via human teleoperation, it is impossible to guarantee that the quality of the data collected on the source and target robots are the same. Therefore, comparing models trained on two separate datasets would not be a fair comparison. However, in simulation, it is possible to create a training dataset on the target robot that is nearly identical to the original training dataset on the source robot. This is achieved by rolling out the same actions on both the source and robot embodiments in scenes with the same initial object positions. In Table 6 we compare the performance between a train-on-source/test-on-source upper bound and a train-on-target/test-on-target upper bound and find no significant difference between them.

| | Source-on-source success rate | Target robot w/ Robotiq | Target-on-target success rate | Target robot w/ Franka | Target-on-target success rate |
|---|---|---|---|---|---|
| Stack | 0.97 | Panda | 0.97 | Panda | 0.94 |
| | | Sawyer | 0.97 | Sawyer | 0.97 |
| | | IIWA | 0.98 | IIWa | 0.95 |
| | | UR5e | 0.98 | UR5e | 0.96 |
| Can | 0.96 | Panda | 0.96 | Panda | 0.94 |
| | | Sawyer | 0.93 | Sawyer | 0.97 |
| | | IIWA | 0.92 | IIWA | 0.96 |
| | | UR5e | 0.96 | UR5e | 0.95 |
| Coffee | 0.95 | Panda | 0.95 | Panda | 0.93 |
| | | Sawyer | 0.91 | Sawyer | 0.95 |
| | | IIWA | 0.90 | IIWA | 0.97 |
| | | UR5e | 0.88 | UR5e | 0.9 |
| Kitchen | 0.95 | Panda | 0.95 | Panda | 0.96 |
| | | Sawyer | 0.99 | Sawyer | 1.0 |
| | | IIWA | 0.93 | IIWA | 0.99 |
| | | UR5e | 0.95 | UR5e | 0.98 |

Table 6: Comparison of two upper-bounds of performance for Shadow. Source-on-Source refers to a policy that is trained on the source embodiment, Panda robot + Robotiq gripper, and evaluated on the source embodiment. Target-on-target refers to a policy that is trained on the target embodiment and evaluated on that same target embodiment.

