# OpenReview forum: "SHADOW: Leveraging Segmentation Masks for Cross-Embodiment Policy Transfer"
_robot-learning.org/CoRL/2024/Conference — CoRL 2024_

### Official Review · Reviewer_PcxW · 2024-07-11
**An excellent paper introducing a simple yet powerful idea**

**Originality:** 4
**Technical Quality:** 5
**Clarity Of Presentation:** 5
**Potential Impact:** 3
**Recommendation:** 4
**Confidence:** 4

**Review:**

######## Strengths ########

1. The main idea of this work, to overlay masks of source and target robots during training end testing, is simple, well explained, and well executed.
2. Connections to prior work are discussed thoroughly and in a way that motivates the proposed changes well
3. Limitations are addressed adequately both throughout the paper and in the Conclusions section
4. The empirical results are quite strong

######## Weaknesses ########

1. The main weakness in this work are well addressed in the manuscript: the need to train a new model for each new embodiment, the assumption that similar strategies can be used for source and target robots to solve the tasks, the need for a calibrated camera
2. One additional strong assumption, which is not really mentioned in the manuscript but is in the video, is that the target robot will be deployed in the same scene as the target robot
3. The choice of "oracle" is perhaps a bit misleading, and could be easily addressed in simulation experiments

######## Arguments ########

This is an excellently executed paper. The idea is very simple: if the main challenge in transferring from one robot to another is their visual appearances confusing vision-based policies, then let's make sure that the appearance during training matches the appearance during evaluation. While other works have also had this insight, this paper also comes up with the cleanest solution among them: masking out the robot pixels and _overlaying the other robot's mask_. This removes the need for any predictive model and simply requires knowledge of the geometry of the robot, which is readily available. And as expected, the results are very strong over a collection of simulated and real-world experiments. I am unaware of any existing approach that has achieved this level of zero-shot cross-embodiment transfer on vision-based policies.

I would appreciate a deeper dive into the assumption, stated in the accompanying video, that the target robot will be deployed in the same scene. If this wasn't the case, would we just require training across multiple scenes + having access to the new scene's camera calibration? Or would the method (and extensions of it) be inapplicable?

In the experimental section, the authors included the performance on the source robot as an "upper-bound". Wouldn't an upper-bound in performance require training no the _target_ robot and evaluating on the same target robot? Because it might in principle be easier to solve tasks with one robot than another. I understand that running this oracle on the real robot might be too expensive because it would require collecting training data on the target robot, but in simulation it should be feasible. Am I missing something?

**Quality Of The Limitations Section:**

3

**Questions For Rebuttal:**

Summarizing the points above, here are the main questions I have for the authors:
- Could the proposed work be extended to operate in new scenes?
- Why choose the source performance as the upper-bound?

######## Additional feedback ########

The following points are provided as feedback to hopefully help better shape the submitted manuscript, but did not impact my recommendation in a major way.

Intro
- "several reasons. First..." --> only 1 reason is given! What about the ability to execute the same motions with a new robot? What if the workspaces or topologies are radically different?

Sec 2
- Ln 72 is the first mention of conditioning on URDFs or aligning action spaces: it comes out of nowhere. At this point I was wondering if the proposed approach would do these two things, and what exactly they meant.
- Nice treatment of related work

Sec 3.1
- The clear statement of assumptions and scope is much appreciated
- The one assumption that I find concerning is that both robots can solve the tasks with similar strategies. It would be good to include a discussion of which tasks do (and do not) fit within this assumption (potentially for a particular pair of robots, or more generally)

Sec 3.2
- I wonder if the authors could instead overlay the robots with some transparency to maintain more of the 3d geometry. I think this would require adding solid backgrounds to both robots where they _don't_ overlap, and a single common solid background to both robots where they do overlap. This should maintain the property that there is no distribution shift.
    - Oh, I see that this was actually addressed in Appendix A.3 by demonstrating that RGB info is not needed. Please include pointers to this and other Appendices wherever relevant to point the reader to the relevant appendices

Sec 4
- "Mirage was already reported to outperform Octo..." --> I recommend giving just a bit more detail on this (e.g., "by margins of at least XXX and up to YYY") to make a more compelling argument. Initially, I was unconvinced until I went to [5] to confirm.
- "The mug cleanup task is especially difficult for the Panda gripper ... whereas the source gripper does not" --> I recommend the authors stick with either Franka or Panda to describe the Panda, instead of moving back-and-forth (e.g., the only time the target gripper is mentioned is in Fig 4 as "Franka gripper")
- How many trials were run per task to obtain statistical significance results?
- The results are strong even on the real robot, but it would've been nice to get a bit of a discussion of the big performance gaps in hexagon and blocks

**Robotics Focus:**

4

**Summary Of Paper:**

The submission studies the setting where there is abundant demonstration data for one robot solving a task, and there is a need to solve the same task using a different robot embodiment. The proposed solution is incredibly simple: mask out the part of the image corresponding to the robot, and overlay a mask of a (rendered) image of the other (target or source, during training and evaluation respectively) robot. This requires access to a CAD model of the robot, knowledge of the kinematics and geometry of the robot, and known camera calibration. Given this information, the training and test input distributions are identical, and as long as the same strategy can be used to solve the problem with the target robot as with the source robot, performance should be close. The authors demonstrate the validity of their approach on both simulated and real-robot experiments.

**Summary Of Recommendation:**

I strongly recommend that this paper is accepted for publication. It is overall an excellent paper. It introduces a simple idea, explains it well, and executes it well. Experiments are well thought out, carried out, and explained. The empirical results are strong. And the main limitations that I can think of are carefully discussed and addressed.

---

### Official Review · Reviewer_ASdD · 2024-07-17

**Originality:** 4
**Technical Quality:** 3
**Clarity Of Presentation:** 4
**Potential Impact:** 3
**Recommendation:** 4
**Confidence:** 2

**Review:**

## Strengths

The method proposed in the paper engages in a meaningful way with the drawbacks of related prior work (mirage) by effectively leveraging a simpler approach which does not rely on difficult video inpainting to achieve a similar objective. The experiments substantiate that this strategy leads to meaningful improvements over the previous method, albeit by requiring some additional privileged information such as the exact calibrated camera position.

Originality) While the method is a relatively simple extension of inpainting methods such as Mirage, I think the idea is still novel and has potential to be expanded substantially in future work.

Clarity) The paper presents the core method well. In general, some additional details should be provided on the concrete details of the pipeline to simplify reproducibility. For example, the data collection method, simulators used (and their errors) etc. should be described in more detail, potentially in the appendix.

The experiments cover a decent range of embodiments (I appreciate the experiment on the deceptively simple gripper change).

Significance) The method sees to perform well and seems simple to set up. However, I discuss some apprehension below.


## Weaknesses

The method requires some privileged ground truth knowledge about the source robot and it's relationship to the target robot, especially the camera position and end-effector position. While the calibration issue is discussed, it is an open question about whether this is a substantial barrier for adoption of the method outside the lab.

Some details on the experiments are unclear, listed below.

## (Undiscussed) Limitations

One discussion I find missing from the paper is the question how different robot embodiments can be before the method starts running into problems. Since the masks of two robots are used, I assume some non trivial similarity in the geometry of the embodiment is required. I would like for the authors to discuss this.

Similarly, if two embodiments need different physical strategies (such as different grips) to achieve a similar task, would the authors still expect their method to generalize well?

Is Shadow robust against other changing details about the scene, such as the ability to transfer from one table top to another? Why or why not? It might be an unrealistic assumption that reference datasets on exactly the same environment are available.

**Quality Of The Limitations Section:**

2

**Questions For Rebuttal:**

## Questions

How were the sample demos collected? Teleop, trained RL agent, scripted policy?

Are there requirements in terms of variety in the training data to enable the method to work?

**Robotics Focus:**

4

**Summary Of Paper:**

The paper presents a method for policy transfer between two robotic embodiments by augmenting an image to include segmentation mask for both the reference and target robot.

**Summary Of Recommendation:**

Overall the method is well motivated and evaluated. I am unsure how much the strong requirements on the camera position limit the applicability of the method.

---

### Official Review · Reviewer_Hf9g · 2024-07-21
**Interesting to see cross embodiment transfer with a simple idea**

**Originality:** 3
**Technical Quality:** 3
**Clarity Of Presentation:** 4
**Potential Impact:** 2
**Recommendation:** 3
**Confidence:** 4

**Review:**

Strength:

1. A simple method that shows successful transfer between robot embodiments
2. Real-world result outperforming baseline (a more complicated impainting method)

Weakness:
1. The method is prone to occlusions
2. The method is reliant on a well-calibrated camera (which the paper reports). It also relies on the robot joint angles to generate a segmentation mask. Hence, this relies on robot joint angles to be well-calibrated (which isn’t reported in the paper).
3. Missing video policy execution on the simulated settings
4. The Limitations section should be in the main text instead of the Appendix
5. I am curious about what would be the main use case for this approach.
    1. For a practical deployment, the robot embodiment is fixed. We can simply do a one-time data collection on the target embodiment.
    2. If this approach can combine data from multiple embodiments, then potentially the use case could be that the method can leverage large internet-scale data for training a policy that generalizes well. However, there is evidence for transfer from only a single source embodiment and we don't have evidence for transfer from multiple source embodiments.

**Quality Of The Limitations Section:**

2

**Questions For Rebuttal:**

Please refer to the weaknesses mentioned above

**Robotics Focus:**

4

**Summary Of Paper:**

This paper presents a simple idea to show cross-embodiment robot behavior transfer between training and test times. Given training data on a source bot, this proposed method is supposed to transfer to any target robot. At training time, segmentation masks of both the source and target robots (virtual) are overlaid on a 3rd person view of the scene which the policy network is conditioned on. Similarly, at evaluation time, the end effectors are aligned and segmentation masks of the source (virtual) and target robots are overlaid on the 3rd person view. The policy works with end effector control and shows successful transfer across embodiments in simulation and the real world.

**Summary Of Recommendation:**

The paper presents real world results which outperforms the baseline. However, it's not clear if the method is scalable and actually useful. Notetheless it's a simple and interesting idea that works in the settings presented in the paper.

---

### Author Rebuttal · Authors · 2024-08-10

Please see updated pdf here and response to all reviewers in the Official Comment.

---

### Decision · Program_Chairs · 2024-09-04

**Decision:**

Accept

**Comment:**

**Paper summary**

This paper presents a method for reusing demonstration data collected on one robot for training another. The proposed method masks out the source robot in the image frame and overlays a mask of the target robot. The evaluation demonstrates how this approach is successful in both simulation and real-world experiments and results in better training than recent baselines.

**Review summary**

Summary of strengths:
+ The proposed method is both simple and effective.
+ The evaluation demonstrates greatly improved results compared to a very recent baseline (Mirage).
+ The experiments demonstrate generalizability across multiple robot platforms.

Summary of weaknesses:
- The proposed method makes several assumptions about the availability and accuracy of data (well-calibrated joint data and camera pose). The evaluation would be improved by additional tests demonstrating how sensitive the proposed method is to these calibrations. **[The rebuttal acknowledges this limitation and poses that it is also a limitation of other SOTA  methods.]**
- The limitations section is present in the supplemental materials, but should be integrated into the main paper. **[This has been addressed in the revision.]**
- The evaluations also assume that the target robot is deployed in the same environment as the source data. The paper should address whether this assumption is necessary for the proposed method to work. **[The rebuttal acknowledges this limitation and proposes strategies for mitigating it.]**

**Response to rebuttal**

The authors have revised the paper to include the requested clarifications and justifications. All authors maintain positive ratings, and one reviewer increased theirs from weak to strong accept.